# Public Knowledge and Attitude towards Vitiligo: A Cross-Sectional Survey in Jordan

**DOI:** 10.3390/ijerph20126183

**Published:** 2023-06-19

**Authors:** Rand Murshidi, Nour Shewaikani, Assem Al Refaei, Balqis Alfreijat, Buthaina Al-Sabri, Mahmoud Abdallat, Muayyad Murshidi, Tala Khamis, Yasmin Al-Dawoud, Zahraa Alattar

**Affiliations:** 1Department of Dermatology, School of Medicine, The University of Jordan, Amman 11942, Jordan; r.murshidi@ju.edu.jo; 2School of Medicine, The University of Jordan, Amman 11942, Jordan; aas0170234@ju.edu.jo (A.A.R.); blq0182125@ju.edu.jo (B.A.); bty2191170@ju.edu.jo (B.A.-S.); tal0182982@ju.edu.jo (T.K.); yas0182840@ju.edu.jo (Y.A.-D.);; 3Department of Neurosurgery, Jordan University Hospital, Amman 11942, Jordan; mahmoud.abdallat@ju.edu.jo; 4Department of Dermatology, The Jordanian Royal Medical Services, Amman 11942, Jordan; muayyadmurshidi2@icloud.com

**Keywords:** vitiligo, knowledge, attitude, skin disorders

## Abstract

Background: Vitiligo is an autoimmune disease affecting approximately 2% of the world’s population. Besides vitiligo cosmetic issues, patients suffer from psychological comorbidities. This results from the stigmatization they encounter from surrounding individuals. Accordingly, the current study was the first to assess Jordanians’ knowledge and attitude toward vitiligo. Methods: Data collection was completed by an online questionnaire consisting of four sections to capture participants’ sociodemographic characteristics, previous exposure, and knowledge and attitude toward vitiligo. The analysis took place through R and RStudio. Results: Of our 994 surveyed participants, only 8.45% and 12.47% had a low level of vitiligo knowledge and negative total attitude score, respectively. Moreover, independent predictors of positive attitudes included younger age (18–30), high school education or lower, hearing about or living with a vitiligo patient, and higher knowledge scores. The highest prevalence of positive attitudes was observed when physicians were the source of knowledge. Conclusion: Some critical misconceptions were identified despite the Jordanian public having sufficient overall knowledge. Furthermore, higher knowledge reflected a higher prevalence of positive attitudes toward the patients. We recommend that future efforts target the public understanding of the nature of the disease and its being non-communicable. Moreover, we emphasize that medical knowledge should be communicated through qualified healthcare providers.

## 1. Introduction

Vitiligo is a chronic autoimmune disease with a prevalence that may reach 2% of the world’s population [1]. The autoimmune activity of the disease renders some areas of the skin depigmented among otherwise healthy-looking skin. Additionally, vitiligo has an incidence that does not correlate with a specific gender or age [1,2]. Historically, the literature has shown that diseases affecting the skin, which makes them easily noticeable to everyone, carry a high stigma toward their patients [3]. Vitiligo is no exception; one study on vitiligo patients in a medical school in Saudi Arabia reported severe impairment to their quality of life (QoL) [4]. Another study showed that the level of impairment could be comparable to cancer patients [5]. Vitiligo patients suffering from anxiety and depression have been consistently implicated in the deterioration of their QoL [6,7]. According to a cross-sectional investigation in Iran, 72% of vitiligo patients had depression that was minimal or mild, whereas 28% and 39.8% had moderate-to-severe depression and anxiety, respectively [8]. A meta-analysis of 15 studies comprising 1176 vitiligo patients indicated a 35.8% prevalence for general anxiety [9]. The public perception of the disease and consequent attitude toward it plays a significant role in patients’ psychological well-being. This was captured in a German study where 90% of patients with vitiligo reported being questioned or approached about their condition, and 24% experienced nasty comments. In addition, 66.7% reported avoidance behavior because of vitiligo or concealing their white spots [10]. Moreover, in Saudi school-age children, misconception about vitiligo included associations with fish/milk foods, calcium deficiency, iron deficiency, infectious origin, relation to chickenpox, being precancerous, and incurability [11]. Whereas one study in south Saudi Arabia found that 83.7% of the respondents had sufficient knowledge about vitiligo [12], another one in the West reported that only 6.9% knew well about the disease [13]. Common misconceptions observed in previous studies include believing that vitiligo is contagious or infectious. Moreover, some believed that witchery and evil spirits might cause vitiligo [14]. In multiple studies investigating attitudes toward patients with vitiligo, participants with sufficient knowledge of vitiligo versus insufficient knowledge reported a higher prevalence of positive attitudes [15,16].

Due to the lack of studies that assessed where Jordanians stand in this respect, our study aimed to be the first to investigate Jordanians’ knowledge about and attitude toward vitiligo. Additionally, the influence of a multitude of factors on their knowledge and attitude was assessed. The significance of this research lies in addressing the misconceptions and negative attitude triggers to direct the efforts in improving vitiligo patients’ quality of life.

## 2. Methods

### 2.1. Study Design

Our cross-sectional study was conducted using a self-administered online questionnaire that was completed by 994 participants from the Jordanian population. A dermatologist and a group of statisticians designed the survey, and the data collection took place in the period between November 2022 and February 2023. The survey included four sections; the first one addressed the participants’ sociodemographic characteristics, including age, gender, educational level, marital status, geographical distribution, household income, employment status, and whether they had a health-related profession. The second section was designed to capture the participants’ previous exposure to vitiligo using four questions: whether they had heard of vitiligo, had been diagnosed with vitiligo, had a partner diagnosed with vitiligo, or lived with someone diagnosed with vitiligo. The third section included 14 questions to assess the participants’ knowledge of the disease, with “Yes,” “Maybe,” and “No” answers. These included questions about the disease’s nature, including it being contagious, hereditary, autoimmune, hygiene related, systemic, food related, lethal, and triggered by psychological distress and magic or witchery. Lastly, the fourth section assessed participants’ attitudes toward vitiligo and constituted 8 statements to evaluate their agreement with them. These included participants’ attitude toward vitiligo patients in situations such as sharing food, becoming friends, shaking hands, hiring decisions, and their intimate relationships among others. Both knowledge and attitude questions were adopted from previous studies [12,15]. In the “knowledge” section, one point was given for a correct response to a question, whereas an incorrect answer and an answer of “Maybe” were given zero points. In the “attitude” section, points were distributed as follows: ”Yes” (+1), “Maybe” (0), and “No” (−1). The summation of responses for each participant computed the total of each score. Besides analyzing the resultant continuous knowledge and attitude scores, participants were classified into three knowledge categories: low knowledge (0–5 points), moderate knowledge (6–10 points), and high knowledge (11–14 points), and into two attitude categories: positive (1–8) and negative (−8–0), to better understand our community’s knowledge about vitiligo.

The minimum required sample size was 385, calculated using the Raosoft sample size calculator, assuming a population size of 11,000,000, a 5% margin of error, a 95% confidence interval, and 50% as the response distribution. Before starting with the data collection, a pilot study of 50 participants was conducted to assess the validity of the questionnaire. Minor modifications were made to the questionnaire, including the grouping of education levels of high school or less in one category. Afterward, data collection started by distributing the questionnaire on social media platforms, including Facebook, WhatsApp, Twitter, and Instagram. The questionnaire was completed anonymously without any identifiers being used to maintain the participants’ privacy throughout the process. A consent form, viewed on the first page of the questionnaire, was obtained from all participants (if the participants agreed to participate, they were asked to click “Start the questionnaire”).

### 2.2. Data Analysis

Data were extracted from the electronic questionnaire into an Excel spreadsheet. Following that, data analysis was conducted through R [17] and Rstudio [18]. The readxl package was utilized to import data into R [19]. Packages that were utilized include tidyverse for data wrangling [20], ggplot2 and ggpubr for plotting [21,22], rstatix for inferential analysis [23], and apaTables for the regression tables [24]. First, descriptive analysis was conducted to generate the counts and proportions of each variable. Then, the Shapiro–Wilk test was used to check the normality of our continuous variables (knowledge and attitude scores). Accordingly, Mann–Whitney and Kruskal–Wallis tests were used to run the inferential analysis of different sociodemographic factors and previous exposure to vitiligo and the knowledge and attitude scores. Finally, multiple linear regression was utilized to identify independent predictors of knowledge and attitude scores. 

## 3. Results

### 3.1. Participant’s Knowledge and Attitude Scores and Sociodemographic Characteristics’ Influence on Them

A total of 994 individuals filled out the self-administered online questionnaire and thus were recruited in the analysis of the current study. The average and standard deviation of knowledge and attitude scores of the total sample were 8.85 ± 2.54 and 4.58 ± 3.26, respectively. Figure 1 shows that 66.7% and 24.85% of the participants had moderate and high knowledge about vitiligo, respectively. Moreover, only 12.47% of the sample had negative total attitude scores. About 61% of the participants were between 16 and 30 years old and showed significantly higher attitudes scores when compared to other age groups (*p* < 0.001 and η2 = 0.105). Females represented 64.39% of our sample and reported significantly higher knowledge scores (*p* < 0.001 and Cohen’s d = 0.13). Furthermore, 55.43% of participants were from Jordan’s capital, Amman. According to the Kruskal–Wallis test, geography did not affect knowledge but was significantly associated with attitude scores (*p* < 0.001 and η2 = 0.031). Specifically, the pairwise Wilcox rank sum test showed that Ammani respondents had higher attitude scores when compared to people from the South governorates of Jordan (*p* < 0.001 and Cohen’s d = 0.201) (Table 1).

Figure 2A shows the proportion of married (40.34%) and single (59.66%) participants. Marital status had a moderate effect, being significantly associated with attitude scores (*p* < 0.001 and Cohen’s d = 0.077; Figure 2C). On the other hand, marital status did not significantly correlate with knowledge scores (*p* = 0.083 and Cohen’s d = 0.002; Figure 2B).

Figure 1 shows the distribution of the participants over three knowledge categories: low, moderate, and high. More than 60% of participants were in the moderate knowledge category. Only 8.8% of participants were in the low knowledge category.

Figure 2 illustrates participants’ marital status and its influence on their knowledge and attitude scores. Figure 2A shows that around 60% of our respondents were single. Figure 2B shows insignificant difference in the knowledge scores in relation to marital status. Figure 2C shows significantly higher attitude scores achieved by single respondents.

In regard to education, Table 2 shows that the majority of respondents had a bachelor’s degree or diploma (66.92%) and paternal and maternal education of high school or less (42.56% and 48.29%, respectively). Personal education level was associated with differential knowledge and attitude scores (*p* < 0.001 and η2 = 0.015, and *p* = 0.034 and η2 = 0.005, respectively). Pairwise analysis showed lower knowledge in participants with personal education of high school or less compared to other groups and a more positive attitude in people with personal or maternal bachelor’s degrees or diplomas compared to participants with higher education (*p* < 0.05). In addition, high paternal education was associated with a more positive attitude when compared to other paternal education groups (*p* < 0.001).

### 3.2. Influence of Previous Exposure to Vitiligo on Knowledge and Attitude Scores

Among surveyed individuals, 93.26% had heard of vitiligo and 2.41% had the disease. Table 3 shows a significant association between being a vitiligo patient, living with a vitiligo patient, and having heard of vitiligo and higher knowledge scores (*p* < 0.05). Attitude scores were higher in people who had heard of vitiligo and lived with a vitiligo patient (*p* < 0.05) Table 3.

### 3.3. Participants’ Sources of Knowledge about Vitiligo and Its Impact on Their Knowledge and Attitude Scores

The results presented in Figure 3A indicate that a substantial proportion of participants had acquired knowledge about vitiligo from family and friends (33.7%) and the internet and social media (26.96%). The Kruskal–Wallis test determined a significant difference in the attitude scores of participants with different knowledge sources (*p* = 0.006 and η2 = 0.013). The subsequent pairwise Wilcoxon rank sum test revealed only one significant difference, in which participants whose source was physicians had a more positive attitude when compared to those who had TV as their knowledge source (*p* < 0.05; Figure 3C). Insignificant differences were observed in knowledge scores (Figure 3B).

Figure 3 illustrates sources of knowledge about vitiligo and their impact on knowledge and attitude scores. Figure 3A shows that most respondents had acquired their knowledge through family and friends, the internet, and social media. Figure 3B shows that knowledge scores were insignificantly higher when the source was physicians. However, Figure 3C shows that physicians as the source of knowledge was associated with a significantly higher prevalence of positive attitudes.

### 3.4. Regression Analysis Identified Independent Predictors of Knowledge and Attitude Scores

The results of multiple linear regression analysis with the knowledge score as the criterion variable revealed the following significant predictors: (1) male sex (beta = −0.5, *p* = 0.025), (2) age more than 50 years (beta = 0.95, *p* = 0.033), (3) health profession (beta = 0.69, *p* = 0.009), (4) participant having vitiligo (beta = 1.54, *p* = 0.045), (5) having heard of vitiligo (beta =1.65, *p* < 0.001), and (6) attitude score (beta = 0.17, *p* < 0.001). The model explained 19.8% of the variance in the knowledge scores, and the F-test validated the fitness (F-statistic: 4.34 on 30 and 529 degrees of freedom, *p* < 0.001) Appendix A.

The results of multiple linear regression analysis with the attitude score as the criterion variable revealed the following significant predictors: (1) ages between 31 and 50 and older than 50 (beta = −1.41, *p* < 0.0001, and beta = −2.33, *p* < 0.001), (2) personal education of high school or less (beta = 1.08, *p* = 0.027), (3) participant living with a vitiligo patient (beta = 1.64, *p* = 0.008), (4) having heard of vitiligo (beta =2.05, *p* < 0.001), and (5) knowledge score (beta = 0.27, *p* < 0.001). The model explained 23.13% of the variance in the attitude scores, and the F-test validated the fitness (F-statistic: 5.31 on 30 and 529 degrees of freedom, *p* < 0.001) Appendix A.

### 3.5. Participants’ Answers to the Questions That Constitute the Knowledge and Attitude Scores

Of our sample, 13.28% reported that vitiligo is a contagious disease. Moreover, over half of the participants (62.17%) denied vitiligo’s autoimmune etiology. When asked about the role of magic and witchery in vitiligo, 16.6% believed that these could trigger vitiligo. For a breakdown of participants’ answers to questions assessing the knowledge score, refer to Appendix A.

Regarding participants’ attitude toward vitiligo, 33.8% disagreed with the statement, “I would marry a vitiligo patient.“ On another note, 85.51% disagreed with the statement, “I would not divorce my partner if they were diagnosed with vitiligo.” Additionally, 84.41% agreed to the statement, “I would hire a vitiligo patient.” For a breakdown of participants’ extent of agreement to statements assessing attitude score, refer to Appendix A.

## 4. Discussion

Vitiligo is a depigmenting autoimmune disease with a prevalence that may reach 2% [1]. Although one may argue that it is a solely cosmetic issue, attention should be paid to the psychosocial stigmatization vitiligo patients face [25,26]. Historically, research has shown that diseases affecting the skin, which makes them easily noticeable to everyone, carry a high stigma toward their patients [3]. Due to the lack of studies that assessed where Jordanians stand in this respect, our study investigated Jordanians’ knowledge about and attitude toward vitiligo. Additionally, the influence of a multitude of factors on their knowledge and attitude was assessed. The significance of this research lies in addressing the misconceptions and negative attitude triggers to direct the efforts in improving vitiligo patients’ quality of life. 

Our sample consisted of 994 participants who were recruited via an online questionnaire. About 64.39% were females and 60.56% and 55.43% were aged between 16 and 30 and lived in the capital, Amman, respectively. When compared to official statistics published by the Department of Statistics at the Jordanian Ministry of Planning and International Cooperation, our sample overrepresents females and people living in Amman, and also received more responses from younger population strata [27].

Only 8.45% and 6.74% of our participants showed a low level of knowledge and had not hear of vitiligo, respectively. Those findings may suggest that the general Jordanian population holds sufficient knowledge about the disease compared to other countries. A similar study in Saudi Arabia reported that 83.7% and 98.2% of participants had adequate knowledge and had heard of vitiligo, respectively [12]. At the same time, a study in South Ethiopia reported 68.3% as the percentage of participants with sufficient vitiligo knowledge [15]. In line came the results of a multinational study, where 66% of the sample had sufficient knowledge, and Arabians had the highest scores [28]. Our results show that among the knowledge items, questions with the fewest correct answers were those about the disease being autoimmune and whether it affects internal organs. Consistently, 78.6% of the Saudi study participants did not know that vitiligo is an autoimmune disease. More importantly, 13.28% of our sample believed that vitiligo is or may be contagious, and 15.5% of the Saudi study agreed with this. Although a small percentage, it could be directly related to the stigma vitiligo patients experience. Another finding that raises concern is that 37.32% of our respondents answered that psychological stress could not trigger vitiligo. If the public becomes more aware that stress contributes to the disease’s progression [29], it may make them cautious with their behaviors. 

In contrast to other studies where social media came at the top [12,30], family and friends were our participants’ most common sources of knowledge. However, social media followed, and its associated attitude scores were among the lowest, especially compared to participants with knowledge acquired from physicians. This was unexpected, given the high usage rates of social media in Jordan and the positive impact it had on public health protection against COVID-19 [31,32]. The association between social media use and poor attitude scores was highlighted to ensure the importance of assertive censorship over social media platforms regarding who can spread medical knowledge. Moreover, interested physicians could be invited to awareness campaigns to speak about the disease and its related psychological issues they come across in clinics. This is supported by an analysis of the accuracy of healthcare source-based and non-healthcare source-based videos on YouTube, where healthcare sources had higher accuracy scores and less misinformation [33]. Unfortunately, a similar and more recent analysis showcased a higher viewer engagement ratio to non-healthcare source-based YouTube videos, along with a low overall quality and accuracy [34].

Our participants’ attitude scores were even more assuring, with only 12.47% having a negative total score. Additionally, participants of young age showed significantly better attitude scores. Findings from other similar studies are consistent with ours [12,35]. This may be viewed optimistically in terms of the idea that future generations will be raised with an emphasis on treating vulnerable people compassionately. Surprisingly, 35.92%, 14.14%, and 5.13% of the respondents declared that they would not marry a vitiligo patient, would let the disease affect their intimate relationship, and would ask for a divorce if their partner developed vitiligo, respectively. Those were the central negative attitudes practiced by our participants, illustrating that even with the overall positive attitude witnessed in this research, it is hard for vitiligo patients to engage in healthy and stable relationships with partners. 

Our results revealed that higher attitude scores correlated with higher knowledge scores. However, education of high school or less was associated with higher attitude scores when compared to higher education. The Saudi study results showed high school and bachelor’s degree levels of education to be associated with higher attitude scores compared to lower education [8]. Considering the independent predictors of lower attitude scores could guide awareness campaigns while determining their targets. 

Implications to our study include the development and establishment of awareness-raising programs and educational campaigns that aim at increasing public awareness about vitiligo, as well as incorporating knowledge about vitiligo into school education, as previous research showed numerous misconceptions in school-age children [11].

The limitations of our study include the use of closed-ended questions that may have restricted participants’ responses, potentially limiting the depth of data obtained. Additionally, the online survey methodology may have introduced sampling bias, as individuals who are less comfortable with technology or lack internet access may have been excluded from the study. Furthermore, our study relied on self-reported answers on questions about attitude and did not address the behavior of participants due to methodological challenges. Recommendations to future researchers involve performing studies on representative samples while also attempting to describe their behaviors in an objective manner.

## 5. Conclusions

Some critical misconceptions were identified despite the Jordanian public having sufficient overall knowledge. Furthermore, higher knowledge reflected a higher prevalence of positive attitudes toward patients. We recommend that future efforts target the public understanding of the nature of the disease and its being non-communicable. Moreover, we emphasize that knowledge should be communicated through qualified healthcare providers, including a multidisciplinary team of medical doctors, pharmacists, nurses, and psychologists, among others. Moreover, our study necessitates the development of educational and accurate online content about vitiligo on global video streaming platforms, including YouTube.

## Figures and Tables

**Figure 1 ijerph-20-06183-f001:**
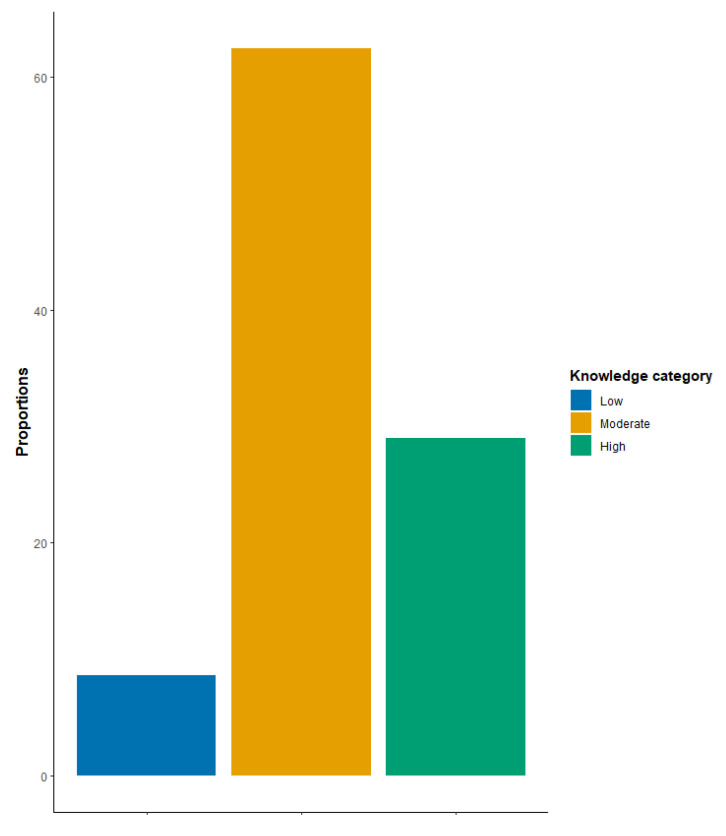
Proportions of the participants’ level of knowledge about vitiligo.

**Figure 2 ijerph-20-06183-f002:**
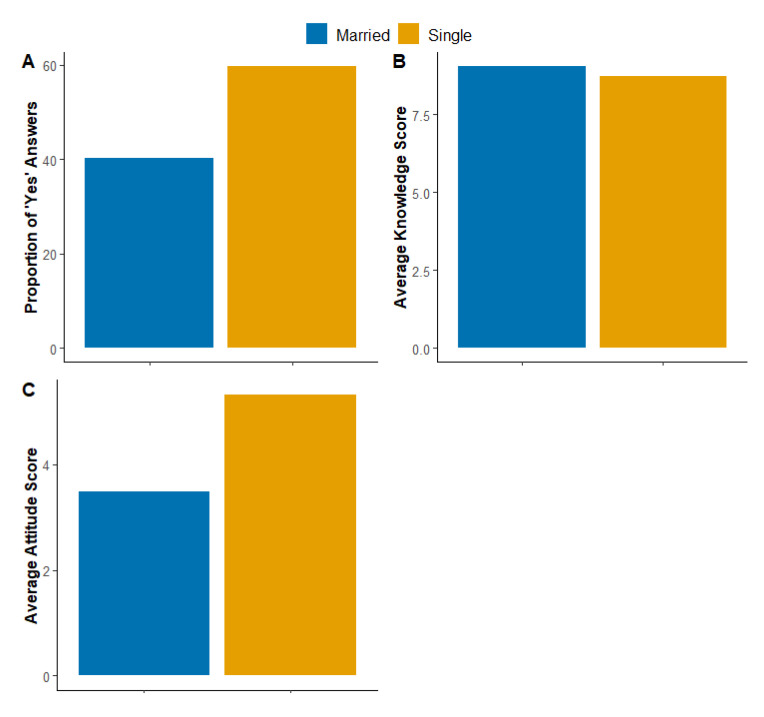
Participants’ marital status and its influence on their knowledge and attitude scores. (**A**) shows the percentages of married and single individuals of our sample. (**B**) shows the average knowledge score according to the marital status of the participants. (**C**) shows the average attitude score according to the marital status of the participants.

**Figure 3 ijerph-20-06183-f003:**
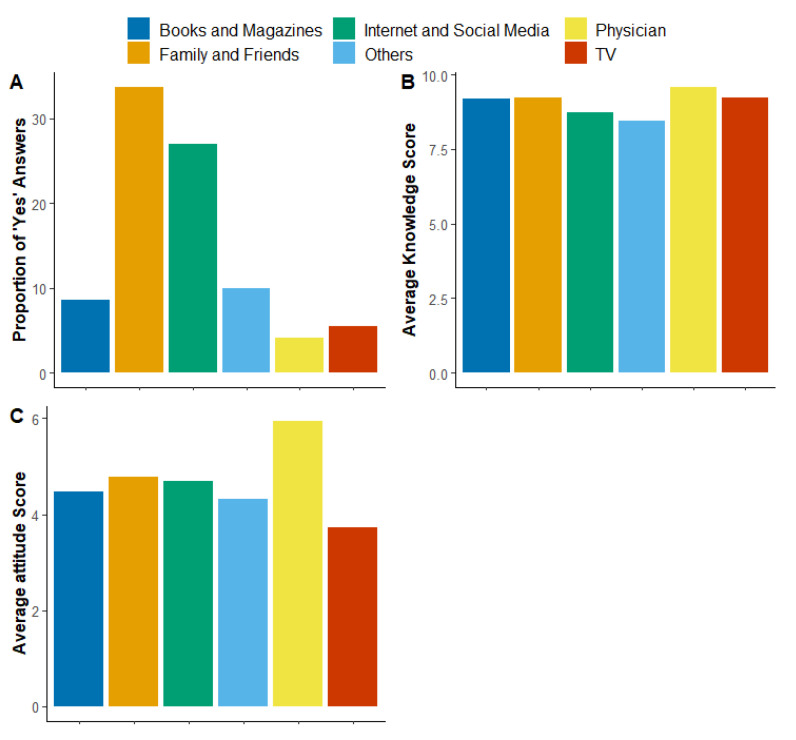
Sources of knowledge about vitiligo and their impact on knowledge and attitude scores. (**A**) shows the percentages of different sources of knowledge about vitiligo. (**B**) shows the impact of the source of knowledge on the average knowledge score. (**C**) shows the impact of the source of knowledge on the average attitude score.

**Table 1 ijerph-20-06183-t001:** Study participants’ background characteristics and knowledge and attitude scores based on some selected characteristics (*n* = 994).

Variables		*n*	%	Knowledge Scores				Attitude Scores			
				Mean	SD	*p*-Value	Effect Size	Mean	SD	*p-*Value	Effect Size
Age group (years)	16–30 years old	602	60.56	8.71	2.59	0.118	0.002	5.42	2.77	**<0.001**	0.105
	31–50 years old	336	33.80	9.06	2.48			3.39	3.5		
	>50 years old	56	5.63	9.04	2.29			2.7	3.66		
Sex	Male	354	35.61	8.31	2.96	**<0.001**	0.13	4.38	3.42	0.237	0.038
	Female	640	64.39	9.15	2.22			4.69	3.16		
	Less than JOD 600	411	41.35	8.45	2.5	**<0.001**	0.02	3.99	3.49	**<0.001**	0.016
Income level	JOD 600–1200	317	31.89	9.00	2.40			4.66	3.18		
	More than JOD 1200	266	26.76	9.08	2.73			5.16	2.98		
Geography	Amman	551	55.43	8.94	2.51	0.679	−0.002	5.05	3.06	**<0.001**	0.031
	Center	117	11.77	8.97	2.22			4.62	2.99		
	North	121	12.17	8.56	2.89			4.12	3.47		
	South	205	20.62	8.69	2.56			3.58	3.54		
Health profession	Yes	451	34.53	9.54	2.03	**<0.001**	0.169	5.23	2.82	**<0.001**	0.12
	No	855	65.47	8.57	2.67			4.32	3.38		
Location	Urban	814	81.89	8.91	2.52	0.056	0.061	479	3.15	**<0.001**	0.132
	Rural	180	18.11	8.56	2.58			3.64	3.56		
Occupation	Unemployed	166	16.7	8.85	2.54	0.234	0.009	4.03	3.45	**<0.001**	0.052
	Employed	466	46.88	8.98	2.47			4.01	3.43		
	Student	362	36.42	8.68	2.62			5.57	2.65		

Note: bold numbers indicate significant difference among categories.

**Table 2 ijerph-20-06183-t002:** Health study of participants’ personal, paternal, and maternal education levels and their influence on their knowledge and attitude scores (*n* = 994).

Variables *n*%		*n*	%	Knowledge Score				Attitude Score			
				Mean	SD	*p-*Value	Effect Size	Mean	SD	*p-*Value	Effect Size
Education level	High school	117	11.77	7.90	2.91	**<0.001**	0.015	4.31	3.89	**0.034**	0.005
	Bachelor’s degree/diploma	747	75.15	8.96	2.40			4.76	3.05		
	Higher education	130	13.08	9.07	2.75			4.79	3.68		
Paternal education level	High school	423	42.56	8.74	2.53	0.211	0.001	3.98	3.4	**<0.001**	0.03
	Bachelor’s degree/diploma	414	41.64	9.01	2.32			4.9	3.13		
	Higher education	157	15.79	8.71	3.04			5.38	2.9		
Maternal education level	High school	480	48.29	8.86	2.49	0.800	−0.002	3.98	3.46	**<0.001**	0.031
	Bachelor’s degree/diploma	440	44.27	8.92	2.35			5.22	2.82		
	Higher education	74	7.44	8.36	3.62			4.69	3.61		

Note: bold numbers indicate significant difference among categories.

**Table 3 ijerph-20-06183-t003:** Health study of participants’ previous exposure to vitiligo and its influence on their knowledge and attitude scores (*n* = 994).

Variables	*n*	%	Knowledge Scores				Attitude Scores			
			Mean	SD	*p*-Value	Effect Size	Mean	SD	*p*-Value	Effect Size
Participant has vitiligo	24	2.41	10.00	1.79	**0.013**	0.079	5.12	3.23	0.305	0.033
Participant lives with a patient with vitiligo	67	6.74	9.99	1.85	**<0.001**	0.13	5.72	2.66	**0.002**	0.099
Participant has heard of vitiligo	927	93.26	9.10	2.18	**<0.001**	0.234	4.77	3.13	**<0.001**	0.184
Participant’s partner has vitiligo	20	2.01	9.15	2.87	0.359	0.039	4.55	4.01	0.218	0.052

Note: bold numbers indicate significant difference among categories.

## Data Availability

The data presented in this study are available upon request from the corresponding author. Respondents’ answers are all reported in tables, and they constitute most of the curated data.

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
