# Peer review of "Public Knowledge and Attitude towards Vitiligo: A Cross-Sectional Survey in Jordan"

_ijerph, 2023, doi:10.3390/ijerph20126183_

Round 1

Reviewer 1 Report

The present work is an epidemiological study regarding vitiligo, a skin pigmentary disease. The introduction, material and methods are very well presented. The results are fine, but minor observations should be addressed, in the results and/or discussion:

1) Normally, vitiligo patients can use makeup to cover the pigmentary alterations. Do the authors have any informations regarding this pratice in Jordan? It is already well-known that this can even be a positive intervention in vitiligo practice.

Li, M., Wang, F., Ding, X., Xu, Q., & Du, J. (2021). Evaluation of the potential interference of camouflage on the treatment of vitiligo: An observer‐blinded self‐controlled study. Dermatologic Therapy34(1), e14545.

2) The present study is very important to highlight different societies, I personally congratulate the authors for the nice discussion with studies from countries outside Europe/US.

3) lines 30-238: how is the situation of social media in Jordan? Please introduce a little about the utilization of social media in your country since this can be a good indicative from the reasons why this is not the principal information source. 

4)lines  244-249: do the authors observed a correlation between the answers regarding relationship with people with vitiligo and sex from the people that answer the questions? Is this more related with masculine public refusing women with vitiligo? If yes, please explore this thema. It is important for the dermatology to understand that any skin alteration represents a social gap. 

5) 250-254: why the Saudi study was different from yours? Please clarify.

6) Conclusions: is it possible to include any other health workers in the conclusion? Although the importance of medical doctors, it is well-known that pharmacits, nurses, psycologists etc can be useful in the imporvement of life quality. In other words, a multidisciplinary team can improve the life quality and information for the population. 

Teovska Mitrevska, N., Eleftheriadou, V., & Guarneri, F. (2012). Quality of life in vitiligo patients. Dermatologic therapy25, S28-S31.

Shourick, J., Ahmed, M., Seneschal, J., Passeron, T., Andreux, N., Qureshi, A., ... & Ezzedine, K. (2021). Development of a shared decision‐making tool in vitiligo: an international study. British Journal of Dermatology185(4), 787-796.

Author Response

Attached file.

Reviewer 2 Report

Your manuscript shows interesting results to the field of vitiligo and I consider it should be published. However, I strongly recommend to add the information about the pilot study with 50 participant to test the questionnaire. It's important to know if you made changes to the questionnaire after the pilot study and at least the sociodemographic characteristics of the participants, mainly their education level. 

One of the limitations of your study is the representativeness of the sample of participants. I agree with you that the electronically acces of the questionnaire could biased the sample. But in the discussion section I suggest to include a commentary about how different was your sample from the general population.

Finally, the figures include in the manuscript show the same information of the text. I recommend not to include them. 

Author Response

Attached file.
